# High sporulation and overexpression of virulence factors in biofilms and reduced susceptibility to vancomycin and linezolid in recurrent *Clostridium* [*Clostridioides*] *difficile* infection isolates

**Laura Tijerina-Rodríguez[1], Licet Villarreal-Treviño[1], Simon D. Baines[2], Rayo Morfín-Otero[3], Adrián Camacho-Ortíz[4], Samantha Flores-Treviño[4], Héctor Maldonado-Garza[4], Eduardo Rodríguez-Noriega[3], Elvira Garza-González[4] ***

**1** Departamento de Microbiología e Inmunología, Facultad de Ciencias Biológicas, Universidad Autónoma de Nuevo León, San Nicolás de los Garza, Mexico, **2** Department of Biological and Environmental Sciences, School of Life and Medical Sciences, University of Hertfordshire, Hatfield, United Kingdom, **3** Instituto de Patología Infecciosa y Experimental, Centro Universitario de Ciencias de la Salud, Universidad de Guadalajara, Hospital Civil de Guadalajara "Fray Antonio Alcalde", Guadalajara, Mexico, **4** Hospital Universitario "Dr. José Eleuterio González", Universidad Autónoma de Nuevo León, Monterrey, Mexico

\* elvira_garza_gzz@yahoo.com

## Abstract

*Clostridium* [*Clostridioides*] *difficile* infection (CDI) is one of the leading causes of diarrhea associated with medical care worldwide, and up to 60% of patients with CDI can develop a recurrent infection (R-CDI). A multi-species microbiota biofilm model of *C. difficile* was designed to evaluate the differences in the production of biofilms, sporulation, susceptibility to drugs, expression of sporulating (*sigH*, *spo0A*), quorum sensing (*agrD$_1$*, and *luxS*), and adhesion-associated (*slpA* and *cwp84*) pathway genes between selected *C. difficile* isolates from R-CDI and non-recurrent patients (NR-CDI). We obtained 102 *C. difficile* isolates from 254 patients with confirmed CDI (66 from NR-CDI and 36 from R-CDI). Most of the isolates were biofilm producers, and most of the strains were ribotype 027 (81.374%, 83/102). Most *C. difficile* isolates were producers of biofilm (100/102), and most were strongly adherent. Sporulation was higher in the R-CDI than in the NR-CDI isolates (p = 0.015). The isolates from R-CDI patients more frequently demonstrated reduced susceptibility to vancomycin than isolates of NR-CDI patients (27.78% [10/36] and 9.09% [6/66], respectively, p = 0.013). The minimum inhibitory concentrations for vancomycin and linezolid against biofilms (BMIC) were up to 100 times and 20 times higher, respectively, than the corresponding planktonic MICs. Expression of *sigH*, *spo0A*, *cwp84*, and *agrD$_1$* was higher in R-CDI than in NR-CDI isolates. Most of the *C. difficile* isolates were producers of biofilms with no correlation with the ribotype. Sporulation was greater in R-CDI than in NR-CDI isolates in the biofilm model of *C. difficile.* The R-CDI isolates more frequently demonstrated reduced susceptibility to vancomycin and linezolid than the NR-CDI isolates in both planktonic cells and biofilm isolates. A higher expression of sporulating pathway (*sigH*, *spo0A*), quorum sensing (*agrD$_1$*),

**Data Availability Statement:** All relevant data are within the paper and its Supporting Information files.

**Funding:** This work was partially supported by Mexico's National Council for Science and Technology, CONACYT, grant 284042. The funder had no role in study design, data collection and analysis, decision to publish, or preparation of the manuscript.

**Competing interests:** The authors have declared that no competing interests exist.

and adhesion-associated (*cwp84*) genes was found in R-CDI than in NR-CDI isolates. All of these factors can have effect on the recurrence of the infection.

## Introduction

*Clostridium* [*Clostridioides*] *difficile* is one of the leading causes of healthcare-associated diarrhea worldwide. Since 2011, cases of *C. difficile* infection (CDI) have increased in the United States, with 453,000 infections and 29,000 deaths [1].

An estimated 20–35% of patients with CDI can develop a recurrent infection (R-CDI) within eight weeks of the first episode, with an incidence of 1,846–37,620 cases/year [2]. The development of R-CDI has been associated with the germination of spores of the strain that produced the initial infection in the colon or with the acquisition of a new strain [3–5].

The pathogenicity of chronic and recurrent infections has been associated with the production of biofilm in some bacterial species [6], and *C. difficile* has been shown to produce organized biofilm communities on abiotic surfaces *in vitro* [7–9]. Furthermore, *C. difficile* has shown to produce biofilms in the presence of other bacteria such as *Finegoldia magna in vitro* and participate in complex gut microbiota biofilms *in vitro* and during infection *in vivo* in a mouse model [10, 11]. Such biofilm formation could protect bacteria from cellular immune responses associated with toxin production and from antibiotics used for the treatment of CDI [10].

*C. difficile* spores have been found within the biofilms in a simulated chemostat gut model [11, 12], suggesting that the accumulation of spores within the biofilm of *C. difficile* could play a role in the development of R-CDI [13]. Lower spore germination rates have been reported in *C. difficile* biofilms than in vegetative cultures [14, 15], which may affect the persistence of infection.

The development of biofilms has been associated with the ability of bacteria to resist antimicrobial agents because they act as a physical barrier and decrease the effective concentration of antimicrobials [8, 10]. Indeed, the biofilms of *C. difficile* have shown 100 times greater resistance to metronidazole and 10 times greater resistance to vancomycin than cells cultured in liquid media [14].

Several factors have been described to have a key role in the formation of *C. difficile* biofilm, such as surface factors like S-layer protein, SlpA (encoded by *slpA*), the cell wall protein Cwp84 (encoded by *cwp84*), and the putative quorum sensing regulator LuxS (encoded by *luxS*) [8, 10, 16]. In addition, the master regulator of sporulation, Spo0A, has shown to determine the biofilm-producing phenotype [7, 14]. Recently, the sigma factor of sporulation, SigH, and the *agr* quorum sensing system, have been shown to regulate metabolism and virulence potential in *C. difficile* [17–20]. Nevertheless, their contributory role in R-CDI development remains unknown.

Therefore, the aim of this study was to evaluate the effect of biofilm production in a range of *C. difficile* ribotypes, their sporulation, antimicrobial susceptibilities, and expression of genes involved in sporulation and biofilm formation in isolates from recurrent and non-recurrent CDI patients.

## Material and methods

### Setting

Patients were recruited for this study who were treated at the following two hospitals in Mexico: The Civil Hospital of Guadalajara, "Fray Antonio Alcalde" a tertiary hospital with

1000 beds in Guadalajara; and the University Hospital "Dr. José Eleuterio González," a tertiary teaching hospital with 500 beds in Monterrey.

## Ethics statement

The local ethics committee (Comité de Ética en Investigación del Antiguo Hospital Civil de Guadalajara "Fray Antonio Alcalde," Jalisco, Mexico) approved this study with reference number 047/16. Informed consent was waived by the Ethics Committee because no intervention was involved and no patients identifying information was included.

## Study population, CDI diagnosis, and classification of CDI

Patients with unexplained diarrhea ($\geq 3$ unformed stools, Bristol scale 5–7) within 24 h were included. For the diagnosis of CDI, fecal samples were collected, and *C. difficile* was investigated by real-time PCR (Cepheid Xpert *C. difficile*/Epi test, Cepheid, Sunnyvale CA). Patients were defined with CDI when patients were diarrheal and PCR was positive.

R-CDI was defined by the reappearance of diarrhea associated with CDI within eight weeks after the completion of antibiotic therapy or the resolution of the initial episode [21]. CDI was classified as NR-CDI when no new episode occurred within eight weeks. Data collected from patients with R-CDI and NR-CDI included epidemiological and clinical data, prior antibiotic therapy, and treatment for CDI. The study was reviewed and approved by the Local Ethics Committee (Approval: 047/16).

## Culture of *C. difficile* and typing of isolates

Fecal specimens were cultured on *C. difficile* agar (Neogen Corporation, MI) with cefoxitin (16 mg/L) and incubated in an anaerobic chamber (10% $CO_2$, 10% $H_2$, and 80% $N_2$) at 37°C for 48 h. Isolates were identified by polymerase chain reaction (PCR) with amplification of the triose phosphate isomerase (*tpi*) gene [22] and by matrix-assisted laser desorption ionization time-of-flight mass spectrometry (MALDI-TOF MS). All isolates were stored at −80°C. The *tcdA*, *tcdB*, *cdtA*, and *cdtB* genes were amplified using a multiplex PCR method [22], and the ribotyping-PCR was performed as previously described [23].

Selected isolates were subjected to ribotyping by capillary electrophoresis at the *C. difficile* Ribotyping Network Reference Laboratory (CDRN) at Leeds Teaching Hospitals NHS Trust (Leeds, United Kingdom).

## *C. difficile* biofilm model

The biofilm of *C. difficile* formation was conducted as reported previously with some modifications. [7, 24] Briefly, each isolate was cultured in brain heart infusion (BHI) broth supplemented with 0.5% yeast extract and 0.1% L-cysteine (BHIS) in 96-well microtiter plates and incubated in anaerobic conditions at 37°C for 7 days. The planktonic cells were removed, and absorbance of planktonic cultures was read at 590 nm using a microtiter plate spectrometer iMarK (Bio-Rad, Hercules, CA, USA). The biofilm was washed with sterile phosphate-buffered saline (PBS) (200 μl), fixed with 2% glutaraldehyde, and washed again with PBS. Then, 1% crystal violet was added and the biofilm was washed six more times with sterile water and de-stained with 30% acetic acid. The optical density was read at 590 nm. The adherence index (AI) was calculated, and the isolates were classified as strong adherents (AI > 1.20), moderate adherents (0.90 < AI <1.20), weak adherents (AI between 0.2 and 0.90) and non-adherents (AI < 0.2) [25]. The assays were performed in triplicate, and only the results with a variation coefficient greater than 20% were accepted.

## Microbiota–*C. difficile* biofilm model

For this study, we designed a biofilm model containing *C. difficile* and species from the intestinal microbiota (*Enterococcus faecalis* ATCC 29212, *E. faecium*, *Lactobacillus casei*, *Lactobacillus rahmnosus*, and *Lactobacillus acidophilus* strains obtained from clinical specimens). Each strain was cultured in BHIS at 37˚C in an anaerobic chamber for 24–48 h and subsequently diluted 1:10 in BHIS broth. A 200-µl mixture of microbiota and *C. difficile* (1:1:1:1:1:1 of each species) were added to each well of a microtiter plate, and the plates were incubated in an anaerobic chamber at 37˚C for 7 days. The determination of the biofilm was carried out according to the biofilm model of *C. difficile* only.

Strains ATCC BAA-1805 (ribotype 027, strong adherent) and ATCC 9689 (ribotype 001, weak adherent) were used as controls in the biofilm assays.

## Antimicrobial susceptibility testing

Antimicrobial susceptibilities were determined by the agar dilution method in selected isolates [26]. Antibiotics used to treat CDI infections, as well as those associated with the development of CDI, were included. The minimum inhibitory concentrations (MICs) were determined for metronidazole (ICN Biomedical, Costa Mesa, CA), vancomycin, linezolid, ciprofloxacin, moxifloxacin, erythromycin, clindamycin, rifampicin, and tetracycline (Sigma-Aldrich). The antimicrobial diluents used and the ranges tested were recommended by the Clinical and Laboratory Standards Institute (CLSI, 2019, M100-S28). An overnight culture in Schaedler broth (Neogen Corporation) of each isolate was inoculated using a multipoint inoculator ($10^4$ colony-forming units (CFU)/spot) on Wilkins-Chalgren agar (Neogen Corporation) [27]. The ATCC 700057 (ribotype 038) was used as a control strain.

Resistance breakpoints were defined according to the CLSI guidelines as follows: moxifloxacin and clindamycin $\geq$ 8 mg/L, tetracycline $\geq$ 16 mg/L, metronidazole $\geq$ 32 mg/L, (CLSI, 2019). The breakpoint for vancomycin was defined according to the European Committee on Antimicrobial Susceptibility Testing (EUCAST, 2019) as greater than 2 mg/L. For antimicrobial agents of which no standard breakpoints to *C. difficile* have yet to be defined, breakpoints were considered as follows: erythromycin $\geq$ 8 mg/L (CLSI, 2013), ciprofloxacin, $\geq$ 8 mg/L, [28] linezolid $\geq$ 16 mg/L [29], and rifampicin $\geq$ 32 mg/L [30].

## Biofilm minimum inhibitory concentration

The BMIC was determined for vancomycin and linezolid. Briefly, the planktonic phase of a 7-day-old biofilm was removed, and the antibiotics were prepared in fresh BHIS. Each concentration (from 512 mg/L to 0.5 mg/L) was aliquoted into microtiter plates (one per concentration), and 200 µl per well were added. The biofilm was resuspended, and the plates were incubated at 37˚C for 48 h in anaerobiosis.

The BMIC was defined as the lowest concentration of an antimicrobial that prevents growth. The ATCC 700057 *C. difficile* strain (ribotype 038) was used as a control.

## Spore count in the biofilm

The 7-day-old biofilm was disrupted with a pipette and resuspended in 100 µl PBS. Serial ten-fold dilutions ($10^{-1}$–$10^{-7}$) were incubated at 65˚C for 20 min to kill the vegetative cells. Both untreated and heat-treated suspensions were streaked on *Clostridium difficile* agar (Neogen Corporation, MI, USA) and incubated at 37˚C for 48 h in anaerobic conditions. Total viable cells and spore counts were determined as CFU/biofilm.

### Quantitative RT-PCR for *spo0A*, *sigH*, *slpA*, *cwp84*, *agrD₁*, and *luxS*

Twenty selected strains (10 from R-CDI patients and 10 from NR-CDI patients) were analyzed. The biofilms incubated for 48 h were washed twice, then the pellet was resuspended in DEPC water (200 µl) and treated with lysozyme (10 mg/ml) (Bio-basic, Ontario, Canada) and proteinase K (Bio-basic). Total RNA was isolated using the Qiagen QIAamp DSP Viral RNA mini kit (QIAGEN, Hilden, Germany). The quantity and quality of the RNA were assessed using a NanoDrop spectrophotometer. The relative quantification of the expression of the RNA transcripts of the *spo0A*, *sigH*, *slpA*, *cwp84*, *agrD₁*, and *luxS* genes normalized to 16S rRNA (*rrs*) was analyzed using the SuperScript III Platinum One-Step kit (Invitrogen, CA, USA).

Standard curves were generated using 5-fold dilutions of ATCC 9689 RNA for each gene to determine the efficiency of the reactions. RNA and diethyl pyrocarbonate (DEPC) water controls were also included. The real-time RT-PCRs were performed in two biological samples in duplicate; 200 ng RNA and 0.5 µl specific primers at 100 nM (listed on Table 1) in a 25-µL reaction volume were used. The Smart Cycler real-time PCR system (Cepheid) was used with cycling conditions as follows: 94˚C for 8 min, then 45 cycles of 94˚C for 30 s; 60˚C for *spo0A*, *sigH* and *cwp84*; 54˚C for *slpA*, *agrD₁* and 30 s for *luxS* and an extension cycle of 72˚C for 25 s. Melting curves were determined to assure that only the expected PCR products had been generated. Relative expression ≥3 was classified as overexpression. Data were normalized and analyzed using the method described by Chang *et al.*, and the ATCC 9689 was used as a calibrator [31].

### Statistical analysis

Reduced susceptibility frequencies from planktonic and biofilm R-CDI cultures were compared with NR-CDI cultures using Pearson's chi-square test and Fisher's Exact test.

Differences in relative expression ratios mean of biofilm cultures from R-CDI and NR-CDI were analyzed using Student's t-test. Non-parametric data were analyzed using the Mann-Whitney test and Spearman rank correlation test.

All statistical analysis were performed using the SPSS software package. A *p* value less than 0.05 was considered to be statistically significant.

## Results

### Study population

In total, 254 patients with CDI (35.29%, female and 64.70%, male; age range, 15–85) were confirmed by PCR, with 102 isolates of *C. difficile* recovered. Patients with R-CDI had more significant exposure to antibiotics before CDI (p = 0.037) than patients with NR-CDI. Fluoro-quinolones and vancomycin were more frequently used in patients with NR-CDI than in those with R-CDI (p = 0.000 and 0.024, respectively). Cephalosporins were more frequently used in patients with R-CDI than in NR-CDI (p = 0.048) (Table 2).

### Culture and ribotyping

Sixty-six isolates (64.70%) of patients with NR-CDI and 36 (35.29%) of patients with R-CDI were obtained. Most of the isolates were found to be ribotype 027 (83/102, 81.37%). The other isolates were found to be ribotypes 003 and 002 (3/102, 2.94% each); 001, 014, 078, 220, and 076 (2/102, 1.96% each one); and 353 (1/102, 0.98%) (Table 3). No significant differences in ribotype distribution between the R-CDI and NR-CDI groups was detected (p = 0.476).

**Table 1. Primer pairs used to amplify the genes studied by real-time RT-PCR.**

| Target gene | 5′ primer | 3′ primer | Source |
|---|---|---|---|
| rrs | GGGAGACTTGAGTGCAGGAG | GTGCCTCAGCGTCAGTTACA | [43] |
| spoA | CTCAAAGCGCAATAAATCTAGGAGC | TTGAGTCTCTTGAACTGGTCTAGG | [44] |
| sigH | GTTGGTAGCAAAAGAAAAAAGTTATGAG | GTACTCTAGTGCTATTTTATCCCCTTCAC | [45] |
| slpA | AATGATAAAGCATTTGTAGTTGGTG | TATTGGAGTAGCATCTCCATC | [43] |
| cwp84 | TGGGCAACTGGTGGAAAATA | TAGTTGCACCTTGTGCCTCA | [43] |
| luxS | GTGTACTTGATGGAGTAAAGGGAGA | TTCTACATCCCATTGGAGATAAGTC | [46] |
| agrD1 | TTTGCTAGCTCATTGGCACTT | GATTGCTGATTTCTTTGGGTACTT | Primer3 software |

## C. difficile biofilm sporulation

In the C. difficile biofilm model, the majority of C. difficile isolates were producers of biofilm (100/102), with 80.55% of R-CDI isolates and 90.90% of NR-CDI isolates being strongly adherent. No differences were detected in biofilm production among the isolates of R-CDI and NR-CDI (AIs geometric mean [GM], 53.87 and 54.06, respectively; p = 0.579).

Sporulation was higher in R-CDI than in NR-CDI isolates (5 $\log_{10}$ CFU/biofilm vs. 3.85 $\log_{10}$ CFU/biofilm; p = 0.015).

## C. difficile–microbiota biofilm and sporulation

In the biofilm of C. difficile–microbiota, no difference was detected in biofilm production among the isolates of R-CDI and NR-CDI (AIs GM, 33.04 and 32.89, respectively; p = 0.677).

**Table 2. Clinical characteristics of patients with R-CDI and NR-CDI.**

| | R-CDI (n = 23) | NR-CDI (n = 31) | p value |
|---|---|---|---|
| Hospitalization | | | |
| Length of stay (mean days, range) | 29.55 (4–124) | 20.86 (4–59) | 0.181 |
| Intensive care unit, n (%) | 6 (26.09) | 10 (32.26) | 0.427 |
| Length of stay in ICU (mean days, range) | 14.77 (2–48) | 12.10 (2–48) | 0.460 |
| Prior antibiotics | | | |
| Any antibiotic, n (%) | 30 (96.77) | 22 (95.65) | 0.675 |
| Length of exposure (mean days, range) | 21.48 (1–100) | 13.52 (1–52) | 0.132 |
| No. of antibiotics (mean) | 3.65 | 2.71 | 0.037* |
| Cephalosporins | 12 (54.54) | 8 (27.59) | 0.048* |
| Clindamycin | 18 (81.82) | 21 (72.41) | 0.329 |
| Macrolides | 21 (95.45) | 27 (91.10) | 0.605 |
| Fluoroquinolones | 10 (45.45) | 27 (91.10) | 0.000* |
| Vancomycin | 9 (40.91) | 21 (72.41) | 0.024* |
| Metronidazole | 19 (86.36) | 25 (86.21) | 0.657 |
| Carbapenems | 14 (63.64) | 17 (58.62) | 0.472 |
| CDI treatment | | | |
| Vancomycin | 19 (86.36) | 24 (80.00) | 0.490 |
| Metronidazole | 13 (59.09) | 19 (63.33) | 0.415 |
| Metronidazole/vancomycin | 10 (45.45) | 16 (56.33) | 0.390 |

Data are no. (%) of patients, unless otherwise noted.

*Significant difference p value <0.05

**Table 3. Ribotype distribution between R-CDI and NR-CDI strains.**

| | Genotype | PCR-Ribotype (n) |
|---|---|---|
| R-CDI (n = 36) | $tcdA^+$, $tcdB^+$, $tcdC$ Δ18+, $cdtA^+/cdtB^+$ | 027 (30) |
| | $tcdA^+$, $tcdB^+$, $tcdC$ Δ18−, $cdtA^-/cdtB^-$ | 003 (1), 001 (1), 076 (1), NT (2) |
| | $tcdA^+$, $tcdB^+$, $tcdC$ Δ18−, $cdtA^+/cdtB^+$ | 353 (1) |
| NR-CDI (n = 66) | $tcdA^+$, $tcdB^+$, $tcdC$ Δ18+, $cdtA^+/cdtB^+$ $tcdA^+$, $tcdB^+$, | 027 (53), |
| | $tcdA^+$, $tcdB^+$, $tcdC$ Δ18−, $cdtA^-/cdtB^-$ | 002 (3), 014 (2) 003 (2), 220 (2), 076 (1), NT (1) |
| | $tcdA^+$, $tcdB^+$, $tcdC$ Δ39+, $cdtA^+/cdtB^+$ | 078 (2) |

No significant difference was detected in sporulation between the R-CDI and NR-CDI isolates (6.21 $\log_{10}$ and 5.54 $\log_{10}$ CFU/biofilm, respectively; p = 0.565).

## Minimum inhibitory concentrations

Drug susceptibility was evaluated on a selection of 65 isolates (26, R-CDI; 39, NR-CDI), and more than 70% of the isolates were resistant to ciprofloxacin (≥8 mg/L), moxifloxacin (≥8 mg/L), erythromycin (≥8 mg/L), clindamycin (≥8), and rifampin (≥32 mg/L). All isolates were susceptible to tetracycline (≤4 mg/L) and metronidazole (≤8 mg/L).

The isolates from R-CDI patients showed a greater reduced susceptibility to vancomycin (>2 mg/L) than the isolates from NR-CDI patients (27.78 and 9.09%, respectively). No other difference was observed between the R-CDI and NR-CDI isolates (Table 4).

## Minimum inhibitory concentrations of the biofilm

In *C. difficile* biofilm isolates without microbiota, a reduced susceptibility to vancomycin was observed in 91.0% (101/102) of isolates and to linezolid in 89.21% (91/102) of isolates.

The BMICs were up to 100-fold higher for vancomycin and 20-fold higher for linezolid than the corresponding MICs. No differences between R-CDI and NR-CDI isolates were observed (Table 5).

## Expression of *spo0A*, *sigH*, *slpA*, *cwp84*, *agrD₁*, and *luxS*

The relative expression of *spo0A*, *sigH*, *cwp84*, and *agrD1* was higher in R-CDI than in NR-CDI isolates (Fig 1). Overexpression of *spo0A* (70%, 7/10), *sigH* (70%, 7/10), *cwp84* (40%, 4/10), and *agrD₁* (70%, 7/10) was higher in R-CDI than in NR-CDI isolates. No significant difference was detected in the expression levels of *slpA* and *luxS* between R-CDI and NR-CDI isolates (Fig 1).

## Discussion

Although some bacterial species that cause recurrent or chronic infections have been studied for their ability to form biofilms *in vivo* and *in vitro* [6], *C. difficile* biofilms have not been widely studied. In the present study, we evaluated biofilm formation by *C. difficile* and detected that most isolates were biofilm producers (strong adherent), independent of the ribotype or whether the strains were isolated from R-CDI or NR-CDI patients. These results reflect those of other previous studies that found no correlation between the ribotype, the strain virulence, or relapse of infection [16].

The expression of quorum sensing regulators and adhesion-associated factors were determined from biofilm culture. *agrD1* and *cwp84* were overexpressed in R-CDI isolates, both of

**Table 4. Antimicrobial susceptibility (mg/L) from R-CDI and NR-CDI strains.**

| Antimicrobial agent | | R-CDI | NR-CDI | *p* value |
|---|---|---|---|---|
| Ciprofloxacin | GM | 111.43 | 61.71 | |
| | Range | 8->128 | 1->128 | |
| | MIC$_{90}$ | >128 | >128 | |
| | Resistant (%) | 96.15 | 89.74 | 0.342 |
| Moxifloxacin | GM | 18.78 | 14.22 | |
| | Range | 1–32 | 1–32 | |
| | MIC$_{90}$ | 32.00 | 32.00 | |
| | Resistant (%) | 92.31 | 87.23 | 0.506 |
| Erythromycin | GM | 190.21 | 87.70 | |
| | Range | 1->128 | 1->128 | |
| | MIC$_{90}$ | >128 | >128 | |
| | Resistant (%) | 81.81 | 95.23 | 0.152 |
| Clindamycin | GM | 150.97 | 66.75 | |
| | Range | 1->128 | 0.5->128 | |
| | MIC$_{90}$ | >128 | >128 | |
| | Resistant (%) | 90.48 | 81.82 | 0.383 |
| Vancomycin | GM | 2.16 | 1.76 | |
| | Range | 1–4 | 0.25–4 | |
| | MIC$_{90}$ | 4.00 | 2.00 | |
| | Resistant (%) | 27.78 | 9.09 | 0.013* |
| Metronidazole | GM | 1.59 | 1.39 | |
| | Range | 0.25–4 | 0.25–4 | |
| | MIC$_{90}$ | 4.00 | 2.00 | |
| | Resistant (%) | 0.00 | 0.00 | NA |
| Linezolid | GM | 3.92 | 4.00 | |
| | Range | 0.5–32 | 0.03–32 | |
| | MIC$_{90}$ | 16.00 | 16.00 | |
| | Resistant (%) | 38.89 | 19.70 | 0.036* |
| Rifampin | GM | 13.55 | 16.02 | |
| | Range | 0.001->128 | 0.002->128 | |
| | MIC$_{90}$ | >128 | 128.00 | |
| | Resistant (%) | 70.59 | 79.41 | 0.484 |
| Tetracycline | GM | 0.26 | 0.16 | |
| | Range | 0.06–8 | 0.06–8 | |
| | MIC$_{90}$ | 4.00 | 0.13 | |
| | Resistant (%) | 0.00 | 0.00 | NA |

Breakpoints were as follows: moxifloxacin and clindamycin ≥8 mg/L, tetracycline ≥16 mg/L, metronidazole ≥32 mg/L according to CLSI (2019); vancomycin >2 mg/L according to EUCAST (2019), erythromycin ≥8 mg/L according to CLSI (2013); ciprofloxacin, ≥8 mg/L [28], linezolid ≥16 mg/L [29] and rifampicin ≥32 mg/L [30].
*Significant difference *p* value <0.05

which were previously shown to regulate colonization, virulence, and relapses in *in vivo* models [8, 16, 18–20]. By contrast, *luxS* and *slpA* expression was similar in R-CDI and NR-CDI isolates. It would be valuable in the future, to analyze the transcription of toxins A and B involving isolates overexpressing *agrD1* and *cwp84* from R-CDI and NR-CDI isolates.

Differences in spore formation in biofilms aged 7–10 days compared to vegetative cultures have been reported previously, including higher viable counts, higher temperature tolerance,

**Table 5. Distribution of MICs and BMICs between R-CDI and NR-CDI strains.**

| | | Vancomycin | | | | Linezolid | | | |
|---|---|---|---|---|---|---|---|---|---|
| | | Range | GM | Resistant (%) | p value | Range | GM | Resistant (%) | p value |
| R-CDI | MIC | 1.00–4.00 | 1.88 | 10/36 (27.78) | 0.000 | 0.50–32.0 | 3.92 | 3/36 (8.33) | 0.000** |
| | BMIC | 2.00–256 | 109.08 | 35/36 (97.22) | | 4.00–256.0 | 87.34 | 29/36 (80.55) | |
| NR-CDI | MIC | 0.25–4.00 | 1.85 | 6/66 (9.09) | 0.000 | 0.03–32.0 | 4.02 | 13/66 (20.0) | 0.000** |
| | BMIC | 2.00–256 | 108.35 | 66/66 (100) | | 2.00–256.0 | 89.63 | 62/66 (95.38) | |

Data are mg/L of an antimicrobial agent unless otherwise noted.

**Significant difference p value <0.01

and pleiomorphic biofilm structures (thin, thick exosporium surrounding the spores in the biofilm) [32, 33]. Further studies to compare germination efficiency with co-germinants in this biofilm model need to be done. In the present study, we analyzed 7-day old biofilms and found that sporulation was greater in the R-CDI strains than in the NR-CDI strains. In addition, sporulation was associated with overexpression of the key regulators of the initial steps of the sporulation pathway, *spo0A* and *sigH*, suggesting their involvement in the overproduction of spores in the biofilm. According to our results, the production of spores can be associated with recurrent CDI isolates.

In the present study, a high proportion of resistance was detected against ciprofloxacin, moxifloxacin, erythromycin, clindamycin, and rifampin, and all isolates were susceptible to tetracycline and metronidazole. No differences were detected between NR-CDI and R-CDI strains for these antimicrobial susceptibilities.

Conversely, lower susceptibility to linezolid was observed in R-CDI strains than in NR-CDI strains, and this result is relevant because linezolid is considered to be a possible drug for treatment with CDI. Interestingly, our study population has no records of exposure to this antimicrobial agent in the 12 weeks before the diagnosis of CDI. The *cfr* gene has been associated with resistance to linezolid and has been detected in *C. difficile* with a MIC up to 16 mg/L [34]. Further studies are underway to clarify the molecular mechanisms associated with this drug resistance.

In our study, the isolates from R-CDI patients showed lower susceptibility to vancomycin (MIC > 2 mg/L) more frequently than isolates from NR-CDI patients (27.78 and 9.09%, respectively); and this result is important given the wide use of vancomycin for treatment of CDI.

High MICs have been reported for moxifloxacin, rifampicin, vancomycin, and clindamycin in ribotypes 001, 017, 027, 176, 078, and 014 [35, 36]. In the present study, high MICs were detected for the same antibiotics in addition to ciprofloxacin and erythromycin. Most of the isolates obtained were ribotype 027. No difference in ribotype was found between the R-CDI and NR-CDI groups. In our population, 027 strains are associated with higher mortality rates and greater probability of R-CDI [37].

*C. difficile* isolates from the present study had high exposure to clindamycin, and this exposure has been associated in prior studies with a high excretion of *C. difficile* spores [38]. Therefore, the high use of clindamycin in our clinical setting of Mexico may be associated with the high sporulation detected.

In a 3-day biofilm model of *C. difficile*, the susceptibility to vancomycin has been reported, with BMICs up to 100 times higher than the corresponding planktonic MICs [8, 39, 40]. In the present study, we confirmed a reduced susceptibility to vancomycin in most strains, with BMICs up to 100 times higher than MICs. Reduced susceptibility of *C. difficile* [27, 41] and

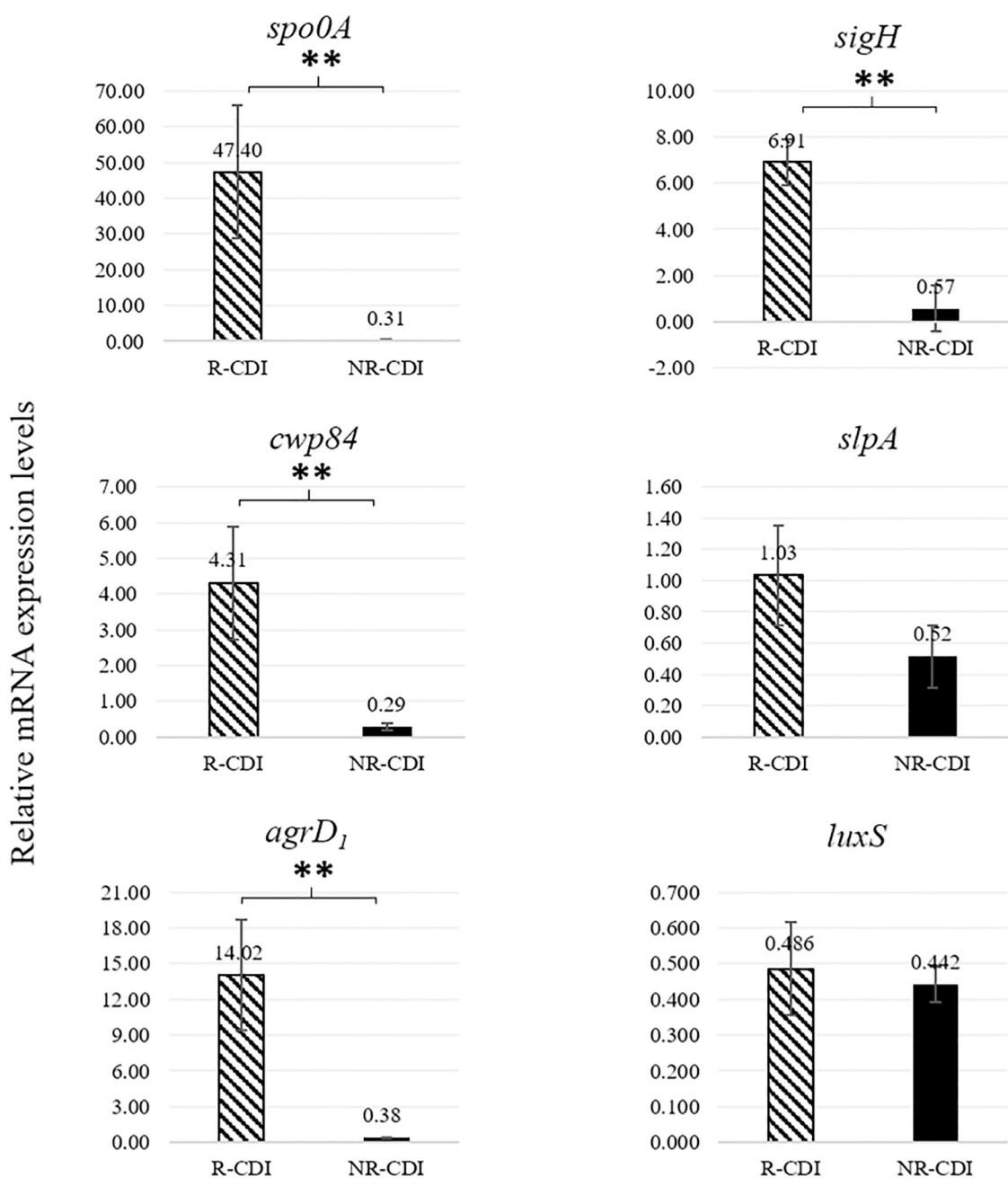

**Fig 1. Expression levels of *spo0A*, *sigH*, *cwp84*, *slpA*, *agrD1*, and *luxS* transcripts between R-CDI and NR-CDI strains.** Relative mRNA transcripts expression means of *spo0A* ($p = 0.003$), *sigH* ($p = 0.007$), *cwp84* ($p = 0.001$), *slpA* ($p = 0.066$), *agrD$_1$* ($p = 0.001$) and *luxS* ($p = 0.400$) from R-CDI and NR-CDI strains. **Significant difference p value <0.01.

BMIC values six times higher than MIC have been reported for metronidazole [14]. Despite our patients having been treated with metronidazole before the diagnosis of CDI, we did not find isolates with reduced susceptibility to this antimicrobial agent in either planktonic or biofilm MIC experiments.

Several risk factors have been described for the development of CDI [42]. In our study, the consumption of cephalosporins and a greater number of previous antibiotics were risk

factors for R-CDI. In addition, patients with NR-CDI more frequently used fluoroquinolones (p = 0.000) and vancomycin (p = 0.024) than those with R-CDI.

The primary limitation of this study was that cultures were not performed on fresh samples. Instead, they were frozen and stored at −20˚C for up to two years with at least two freeze-thaw cycles, which could explain the low recovery of *C. difficile.*

In conclusion, the sporulation and overexpression of *sigH*, *spo0A*, and *agrD₁* were greater in R-CDI than in NR-CDI isolates. The R-CDI isolates had more reduced susceptibility to vancomycin and linezolid than the NR-CDI isolates in both planktonic cells and biofilm isolates. These factors may affect the recurrence of the infection because a greater sporulation in the protected biofilm may facilitate less spore washout from the gut and a higher likelihood of *C. difficile* remaining after CDI therapy has ceased.

## Author Contributions

**Conceptualization:** Laura Tijerina-Rodríguez, Licet Villarreal-Treviño, Simon D. Baines, Adrián Camacho-Ortíz, Elvira Garza-González.

**Data curation:** Laura Tijerina-Rodríguez, Rayo Morfín-Otero, Adrián Camacho-Ortíz, Eduardo Rodríguez-Noriega, Elvira Garza-González.

**Formal analysis:** Laura Tijerina-Rodríguez, Simon D. Baines, Elvira Garza-González.

**Funding acquisition:** Laura Tijerina-Rodríguez, Licet Villarreal-Treviño, Rayo Morfín-Otero, Adrián Camacho-Ortíz, Samantha Flores-Treviño, Héctor Maldonado-Garza, Eduardo Rodríguez-Noriega, Elvira Garza-González.

**Investigation:** Laura Tijerina-Rodríguez, Simon D. Baines, Rayo Morfín-Otero, Adrián Camacho-Ortíz, Eduardo Rodríguez-Noriega, Elvira Garza-González.

**Methodology:** Laura Tijerina-Rodríguez, Simon D. Baines, Elvira Garza-González.

**Project administration:** Laura Tijerina-Rodríguez, Licet Villarreal-Treviño, Rayo Morfín-Otero, Adrián Camacho-Ortíz, Samantha Flores-Treviño, Elvira Garza-González.

**Resources:** Licet Villarreal-Treviño, Rayo Morfín-Otero, Adrián Camacho-Ortíz, Samantha Flores-Treviño, Héctor Maldonado-Garza, Eduardo Rodríguez-Noriega, Elvira Garza-González.

**Software:** Samantha Flores-Treviño.

**Supervision:** Simon D. Baines, Elvira Garza-González.

**Validation:** Laura Tijerina-Rodríguez.

**Visualization:** Laura Tijerina-Rodríguez.

**Writing – original draft:** Laura Tijerina-Rodríguez, Elvira Garza-González.

**Writing – review & editing:** Laura Tijerina-Rodríguez, Licet Villarreal-Treviño, Simon D. Baines, Rayo Morfín-Otero, Adrián Camacho-Ortíz, Samantha Flores-Treviño, Héctor Maldonado-Garza, Eduardo Rodríguez-Noriega, Elvira Garza-González.

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
