## [Decision Letter · Decision Letter 0]

2 Jul 2019

PONE-D-19-15391

High sporulation and overexpression of virulence factors in biofilms and reduced susceptibility to vancomycin and linezolid in recurrent Clostridium [Clostridioides] difficile infection isolates

PLOS ONE

Dear Dr. Garza-González,

Thank you for submitting your manuscript to PLOS ONE. After careful consideration, we feel that it has merit but does not fully meet PLOS ONE’s publication criteria as it currently stands. Therefore, we invite you to submit a revised version of the manuscript that addresses the points raised during the review process.

Your manuscript has been reviewed by an expert in your field. Based on the reviewer's comments, a major revision is needed. I hope that the reviewer's comments are useful to you.

We would appreciate receiving your revised manuscript by 4 weeks. To enhance the reproducibility of your results, we recommend that if applicable you deposit your laboratory protocols in protocols.io, where a protocol can be assigned its own identifier (DOI) such that it can be cited independently in the future. For instructions see: http://journals.plos.org/plosone/s/submission-guidelines#loc-laboratory-protocols

We look forward to receiving your revised manuscript.

Kind regards,

Yung-Fu Chang

Academic Editor

PLOS ONE

**Journal Requirements:**

**Comments to the Author**

1. Is the manuscript technically sound, and do the data support the conclusions?

Reviewer #1: Yes

2. Has the statistical analysis been performed appropriately and rigorously? 

Reviewer #1: Yes

3. Have the authors made all data underlying the findings in their manuscript fully available?

Reviewer #1: Yes

4. Is the manuscript presented in an intelligible fashion and written in standard English?

Reviewer #1: Yes

5. Review Comments to the Author

Reviewer #1: Authors compared C diff isolates from patients with NR-CDI (non-recurrent CDI) and R-CDI (recurrent CDI) in sporulation, resistance to vancomycin and linezolid, biofilm formation in the presence or absence of other bacteria, biofilm resistance to vancomycin and linezolid, and transcription of factors related in sporulation, biofilm formation, adhesion (spo0A, sigH, slpA, cwp84, agrD1 and luxS). They found that R-CDI isolates had hogher sporulation rates, were more resistant to antibiotics, had higher transcription of sigH, spo0A, agrD1, and Cwp84. The results are interesting to understand to R-CDI and NR-CDI. Howerer, there are issues need to be addressed:

1) line 300: minumum inhibitory concentraion of the biofilm was determined in C didd bioflm model or mocrobiota-Cdiff biofim model?

2) It would be more informative to check transcription of toxins, germination of spores, and adhesionand biofilm formation capability.

6. PLOS authors have the option to publish the peer review history of their article (what does this mean?). If published, this will include your full peer review and any attached files.

Reviewer #1: No

---

## [Author Response · Author response to Decision Letter 0]

3 Jul 2019

1) line 300: minimum inhibitory concentration of the biofilm was determined in C diff bioflm model or mocrobiota-Cdiff biofim model?

Response: The minimum inhibitory concentrations were determined from C. difficile biofilm without microbiota. The point was clarified in the new manuscript in line 301.

2) It would be more informative to check transcription of toxins, germination of spores, and adhesion and biofilm formation capability.

Response: Regarding the adhesion and biofilm formation, we calculated the adherence capability as adherence index for each isolate producing biofilm, and there was no difference between R-CDI and NR-CDI. This information is included in the manuscript. 

We added transcription of toxins as perspectives of the study (line 336). However, we expect a low impact because toxin transcription has shown to be ribotype-dependent and our population was mostly infected with 027 ribotype in either R-CDI or NR-CDI isolates.

We also added germination of spores as perspective (line 330) because our data support the role of sporulation pathway genes (sigH, spo0A), in the development of recurrent CDI.

---

## [Decision Letter · Decision Letter 1]

22 Jul 2019

High sporulation and overexpression of virulence factors in biofilms and reduced susceptibility to vancomycin and linezolid in recurrent Clostridium [Clostridioides] difficile infection isolates.

PONE-D-19-15391R1

Dear Dr. Garza-González,

We are pleased to inform you that your manuscript has been judged scientifically suitable for publication and will be formally accepted for publication once it complies with all outstanding technical requirements.

With kind regards,

Yung-Fu Chang

Academic Editor

PLOS ONE

Additional Editor Comments (optional):

Reviewers' comments:

Reviewer's Responses to Questions

**Comments to the Author**

1. If the authors have adequately addressed your comments raised in a previous round of review and you feel that this manuscript is now acceptable for publication, you may indicate that here to bypass the “Comments to the Author” section, enter your conflict of interest statement in the “Confidential to Editor” section, and submit your "Accept" recommendation.

Reviewer #1: All comments have been addressed

2. Is the manuscript technically sound, and do the data support the conclusions?

Reviewer #1: Yes

3. Has the statistical analysis been performed appropriately and rigorously? 

Reviewer #1: Yes

4. Have the authors made all data underlying the findings in their manuscript fully available?

Reviewer #1: Yes

5. Is the manuscript presented in an intelligible fashion and written in standard English?

Reviewer #1: Yes

6. Review Comments to the Author

Reviewer #1: all comments have been addressed. The manuscript is technically sound, with appropriate statistics. Well written with clarity and simplicity.

7. PLOS authors have the option to publish the peer review history of their article (what does this mean?). If published, this will include your full peer review and any attached files.

Reviewer #1: No

---

## [Editor Report · Acceptance letter]

23 Jul 2019

PONE-D-19-15391R1 

High sporulation and overexpression of virulence factors in biofilms and reduced susceptibility to vancomycin and linezolid in recurrent *Clostridium* [*Clostridioides*] *difficile* infection isolates. 

Dear Dr. Garza-González:

I am pleased to inform you that your manuscript has been deemed suitable for publication in PLOS ONE. Congratulations! Your manuscript is now with our production department. 

With kind regards,

on behalf of

Dr. Yung-Fu Chang 

Academic Editor

PLOS ONE